# Treatment and Prognosis of Oral Cancer Patients with Confirmed Contralateral Neck Metastasis: A Multicenter Retrospective Analysis

**DOI:** 10.3390/ijerph19159229

**Published:** 2022-07-28

**Authors:** Yuki Sakamoto, Mitsunobu Otsuru, Takumi Hasegawa, Masaya Akashi, Shin-ichi Yamada, Hiroshi Kurita, Masaya Okura, Nobuhiro Yamakawa, Tadaaki Kirita, Souichi Yanamoto, Masahiro Umeda, Yuka Kojima

**Affiliations:** 1Department of Dentistry and Oral Surgery, Kansai Medical University Medical Center, Moriguchi 570-8507, Japan; s.yukioutdoor@gmail.com; 2Department of Clinical Oral Oncology, Nagasaki University Graduate School of Biomedical Sciences, Nagasaki 852-8588, Japan; mumeda@nagasaki-u.ac.jp; 3Department of Oral and Maxillofacial Surgery, Kobe University Graduate School of Medicine, Kobe 652-0032, Japan; hasetaku@med.kobe-u.ac.jp (T.H.); akashimasaya0105@gmail.com (M.A.); 4Department of Oral and Maxillofacial Surgery, Faculty of Medicine, Academic Assembly, University of Toyama, Toyama 930-8555, Japan; shinshin@med.u-toyama.ac.jp; 5Department of Dentistry and Oral Surgery, Shinshu University School of Medicine, Matsumoto 390-8621, Japan; hkurita@shinshu-u.ac.jp; 6Department of Oral and Maxillofacial Surgery, Saiseikai Matsusaka General Hospital, Matsusaka 515-8557, Japan; okura@dent.osaka-u.ac.jp; 7Department of Oral and Maxillofacial Surgery, Nara Medical University, Kashihara 634-8521, Japan; yamanobu@naramed-u.ac.jp (N.Y.); tkirita@naramed-u.ac.jp (T.K.); 8Department of Oral Oncology, Hiroshima University, Hiroshima 739-8511, Japan; syana@hiroshima-u.ac.jp; 9Department of Dentistry and Oral Surgery, Kansai Medical University, Hirakata 573-1191, Japan; kojimayk@hirakata.kmu.ac.jp

**Keywords:** mouth neoplasms, lymphatic metastasis, prognosis, neck dissection, risk factor

## Abstract

The prognosis of oral cancer that has metastasized to the contralateral cervical lymph nodes is poor, although the appropriate treatment method has not been established because of its rarity. A multicenter retrospective study on the treatment and prognosis of pN2c oral cancer patients was conducted. We investigated the treatment and prognosis of 62 pN2c patients out of 388 pN+ patients with oral squamous cell carcinomas. Statistical analysis was performed on the various factors with overall survival (OS) and disease specific survival (DSS). In multivariate cox regression analysis, advanced T stage was significantly correlated with poor OS (*p* = 0.011) and DSS (*p* = 0.023) of patients with pN2c neck. In pN2c patients, OS, DSS, and neck control was not different between those undergoing ipsilateral neck dissection initially and those undergoing bilateral neck dissection. Thus, contralateral elective neck dissection is not recommended. The most important risk factor for prognosis in pN2c oral cancer patients is advanced T stage. No evidence was found to recommend contralateral elective neck dissection in clinically N1/2b patients. Therefore, the indication for contralateral elective neck dissection in N1/2b patients should be carefully determined in consideration of individual conditions.

## 1. Introduction

The number of oral cancers ranks 18th in the world, with 2% of all cancers, and 14th in the number of deaths, and 1.9% of all cancers [1]. Excluding some areas where oral cancer occurs frequently, such as South Asia [2], oral cancer is rare, including in Japan. Therefore, there are not many studies with a high level of evidence using a large number of cases of oral cancer, and the current situation is that treatment methods vary on each hospital. It is well known that cervical lymph node metastasis is very important prognostic factor for oral cancer. The most reliable treatment for cervical lymph node metastasis is neck dissection. Several recent randomized controlled studies [3,4], and observational study using propensity score matching analysis [5] have shown recommendations for elective neck dissection for N0 tongue cancer. On the other hand, there are many reports that elective neck dissection lowers the patient’s quality of life (QOL) and that the wait-and-see policy can be adequately adopted with sufficient follow-up [6,7,8,9]. In Japan, many large facilities specializing in oral cancer exist in a small land area and the medical insurance system is generous. Therefore, it is possible to perform strict follow-up using CT, MRI, ultrasound, and other imaging tests at short intervals.

Some reports suggest that the frequency of contralateral or bilateral neck metastasis of oral cancer is as rare as 0.9% and others as high as 24% [8,10,11,12,13,14,15,16,17]. Therefore, in Japan, there is no consensus of an appropriate neck dissection for contralateral or bilateral cervical metastasis, and there are no indication criteria for contralateral elective neck dissection when the ipsilateral N+, but contralateral N− case. The objectives of this multicenter observational study were to investigate (1) factors predicting contralateral metastasis, (2) prognosis of pN2c cases compared to pN1 and pN2b, and (3) the difference between contralateral elective neck dissection and a contralateral wait-and-see policy.

## 2. Materials and Methods

### 2.1. Study Design and Patients

This is a retrospective observational study. The subjects were patients with oral squamous cell cancer with histologically proven neck metastases who underwent neck dissection at Department of Oral and Maxillofacial Surgery of Nagasaki University Hospital, Kobe University Hospital, Shinshu University Hospital, Osaka University Dental Hospital, and Nara Medical University from 2000 to 2017. The prognosis was confirmed as of January 2021.

### 2.2. Ethical Approval

The study protocol conformed to the ethical guidelines of the Declaration of Helsinki and the Ethical Guidelines for Medical and Health Research involving Human Subjects by the Ministry of Health, Labor, and Welfare of Japan. The study was approved by the Institutional Review Board (IRB) of the Nagasaki University Hospital. Japanese law does not require individual informed consent from participants in non-invasive observational trials such as the present study. Therefore, the need for informed consent was waived according to the IRB comments. As this was a retrospective study, patient identifiable information was removed, and the research plan was published on the homepages of the participating hospitals websites, along with an opt-out option in accordance with the IRB of Nagasaki University Hospital instructions. As this was a retrospective observational study, it was not registered.

### 2.3. Data Examined

The following information was retrieved from the medical records: age, gender, primary site, differentiation, tumor site superficial and deep (whether invade to the midline), clinical T stage, clinical N stage, neoadjuvant chemo-/radiation therapy, postoperative chemo-/radiation therapy, number of positive lymph node metastases in ipsilateral/ contralateral neck, extranodal spread in the ipsilateral/contralateral neck, level of metastasis in the ipsilateral/contralateral neck (level 1–3/4–5), and treatment outcomes. Clinical T stage and N stage were classified according to the TNM Classification of Malignant Tumours, 7th edition of UICC, and pathological N stage was classified by histological examination of the maximal cut surface of the resected lymph nodes. Endpoints were overall survival (OS) and disease specific survival (DSS) in each of pN1, pN2b, and pN2c, and relationship between dissection time and PS/DSS in those with contralateral neck metastasis.

### 2.4. Statistical Analysis

All statistical analyses were performed using SPSS software (version 26.0; Japan IBM Co., Ltd., Tokyo, Japan), and a two-tailed *p*-value ˂ 0.05 was considered significant. The survival curve is calculated by the Kaplan–Meier method and analyzed by the log-rank test. Factors related to prognosis were analyzed by Cox proportional hazards model using variables with a *p* value less than 0.1 in the log-rank test as covariates. Proportional hazardousness was determined by drawing a log-minus-log curve.

## 3. Results

### 3.1. Patient Characteristics

A total of 388 patients were enrolled including late metastasis, of whom 109 patients were pN1,217 were pN2b, and 62 were pN2c. Demographic characteristics of patients were as shown in Table 1. The subjects consisted of 256 males and 132 females, with an average age of 65.4 years old. The most common primary site was tongue, followed by lower gingiva, upper gingiva/palate, floor of the mouth, buccal mucosa, etc.

Regarding the location of the primary tumor in N2c patients, 15 patients had a unilateral tumor, while 47 had a primary tumor that invaded the opposite side beyond the midline.

### 3.2. Treatment Outcome of Patients with Neck Metastasis

The five-year OS in patients with pN1, pN2b, and pN2C was 61.0%, 47.2%, and 31.2%, respectively. There was a significant difference between OS of pN1 and pN2b (*p* = 0.005), and pN1 and pN2c (*p* < 0.001), whereas difference between OS of pN2b and pN2c was not significant (*p* = 0.083). The five-year DSS in patients with pN1, pN2b, and pN2C was 69.1%, 55.7%, and 45.5%, respectively. Like OS, there were significant differences between DSS of pN1 and pN2b (*p* = 0.008), and between pN1 and pN2c (*p* = 0.001), although no significant difference between DSS of pN2b and pN2c (*p* = 0.189). There was no apparent difference in the causes of death among pN1, pN2b, and pN2c (Figure 1).

### 3.3. Factors Related to Treatment Outcome of Patients with Contralateral Neck Metastasis

Table 2 shows the results of log-rank test of factors related to the prognosis of pN2C. OS was significantly lower in patients with advanced T stage (*p* = 0.031), extranodal spread in the ipsilateral neck (*p* = 0.002), and level 4–5 metastasis in the ipsilateral neck (*p* = 0.032). DSS was significantly lower in those with advanced T stage (*p* = 0.007), extranodal spread in the ipsilateral neck (*p* = 0.006), and level 4–5 metastasis in the ipsilateral neck (*p* = 0.004). However, timing of surgery of the contralateral neck (initially ipsilateral and consequently contralateral neck/initially bilateral necks) did not become a significant factor related to the prognosis of pN2c (OS: *p* = 0.500, DSS: *p*= 0.406).

Table 3 shows multivariate Cox regression analysis of prognostic factors. Advanced T stage was significantly correlated with poor OS (HR, 2.753; 95% CI, 1.344–5.1637; *p* = 0.006), and with poor DSS (HR, 3.883; 95% CI, 1.569–9.609; *p* = 0.003). Furthermore, level 4–5 metastasis significantly correlated with poor OS (HR, 2.205; 95% CI, 1.008–4.821; *p* = 0.048) and DSS (HR, 2.623; 95% CI, 1.162–3.988; *p* = 0.020).

The survival rate of pN2c by neck dissection method is shown in Figure 2. There was no statistical difference between the prognosis of those undergoing ipsilateral neck dissection initially and those undergoing bilateral dissection. When excluding clinical N2c cases, the prognosis of patients with bilateral dissection was significantly worse in OS and DSS than those with ipsilateral dissection cases. This is thought to be due to the fact that there were many cases of advanced T stage in patients undergoing bilateral dissection despite clinically N− in the contralateral neck. In patients with bilateral neck dissection, T3–4 tumor was seen in 11 of 13 (85%) and six of 13 (46%) resulting in local failure, while T3–4 was seen in 10 of 23 (43%), and local failure was observed only in three of 23 (13%) in patients undergoing ipsilateral dissection (Figure 3).

### 3.4. Subgroup Analysis of Patients with Carcinoma of the Tongue or the Floor of the Mouth

Since the rate of metastasis to the contralateral/bilateral necks of cancer of the tongue or the floor of the mouth was higher than that of other cancers (19.3% vs. 12.2%), a subgroup analysis was performed. In a univariate analysis, poor OS was significantly correlated with extranodal spread and level 4–5 metastasis in the ipsilateral neck, while poor DSS was significantly correlated with advanced T stage and level 4–5 metastasis in the ipsilateral neck in patients with tongue or floor of the mouth cancer having pN2c necks (Table 4). Multivariate analysis showed level 4–5 metastasis in the ipsilateral neck as a significant risk factor for poor OS and DSS (Table 5). Figure 4 shows OS and DSS curves in the tongue or the floor of the mouth cancer patients with pN2c necks. No apparent differences in prognosis were seen between methods of neck dissection, although neck failure was slightly more common in the bilateral dissection group.

## 4. Discussion

This retrospective study showed no evidence to recommend contralateral elective neck dissection in clinically N1/2b oral cancer patients.

The question of whether or not elective neck dissection is necessary for clinical N0 patients with oral cancer remains controversial. Vandenbrouck et al. [11] reported the first randomized controlled trial for patients with T1-3N0 oral cancer in 1980, concluding that survival rate of the elective dissection group and wait-and-see policy group did not differ. On the other hand, Fakih et al. reported in 1989 that a randomized controlled trial showed the disease-free survival rate of N0 tongue cancer was higher in the elective dissection group than that in the wait-and-see group [12]. Subsequent, in all the three randomized controlled trials conducted in the 1990s [13], 2000s [14], and 2010s [4], patients with N0 tongue cancer with elective neck dissection for had a better prognosis than those with a wait-and-see policy. Several recent systematic reviews have also recommended elective neck dissection for N0 tongue cancer [15,16]. However, the treatment results for oral cancer in Japan may differ significantly from these papers. The five-year survival rates of 1234 patients in the Otsuru’s paper [5] were 85.5% in the elective neck dissection group and 90.2% in the wait-and-see group, while the three-year survival rates of 496 patients in the D’Cruz’s paper [4] were 80.0% in the elective neck dissection group and 67.5% in the wait-and-see group. In Japan, there are many facilities specializing in oral cancer treatment in a small land area, which makes it easy for patients to access. Furthermore, the medical insurance system in Japan allows frequent follow-up of cervical lymph nodes using modalities, such as CT and MR. Under these circumstances, we think that it may be possible to detect late neck metastasis at an early stage even with a wait-and-see policy, and a good prognosis may be expected in Japan.

It has been reported that contralateral and bilateral neck metastasis is accounted for 0.9–34.7% in oral squamous cell carcinoma [8,12,18,19]. In this study as well, the metastasis rate to the contralateral neck among pN+ patients was 15.9% (62/388), which was similar to the previous reports. Due to the small number of cases of contralateral metastasis, the factors related to the contralateral metastasis, appropriate treatment methods and prognosis of pN2c patients have not been clarified. Kowalski et al. [8], Donaduzzi et al. [17], and Singh et al. [20] reported advanced T stage and primary tumor crossing the midline were related to the contralateral neck metastasis. Kurita et al. [10] and Habib et al. [21] reported that tumor cell differentiation and number of positive nodes became risk factors for contralateral metastasis, while delayed diagnosis, nerve invasion [22], and depth of invasion [23] were also described as factors related to the contralateral neck metastasis. However, these are conclusions based on the observation of a small number of cases and have not been statistically analyzed.

When bilateral neck metastases are clinically observed, bilateral neck dissection is performed, but the treatment strategy of the contralateral neck in clinically unilateral metastases is controversial. Lim et al. [24] reported that contralateral elective neck dissection did not improve survival in an early-stage tongue cancer, so a close follow-up was recommended for contralateral neck in clinical unilateral N+ patients. Klingelhoffe et al. [25] stated that patients with initial bilateral neck dissection had more cervical recurrence than the ipsilateral dissection alone, and thus contralateral elective neck dissection was not required. Singh et al. [20] concluded that contralateral elective dissection was not recommended since it could cause dysfunction, such as long-term chronic neuropathic pain. Habib et al. [21] stated that elective dissection was not necessary because patients with contralateral metastases often died from local recurrence or distant metastases, and contralateral control did not improve survival.

In contrast, Nobis et al. [26] evaluated two elective neck dissection variations, unilateral and bilateral, in patients with tongue cancer to determine the optimal extent. A total of 150 patients were identified, 105 receiving unilateral and 45 bilateral elective neck dissection. In conclusion, they recommend bilateral neck dissection because of advantages with regard to oncologic safety and esthetic outcome. Koo et al. [27] also reported that tumors with metastases on the ipsilateral neck and more than T3 or beyond midline tumor had a high risk of contralateral metastases and required bilateral elective neck dissection or adjuvant radiation therapy. However, neither report strongly recommended contralateral elective neck dissection, and they stated that treatment should be selected with an emphasis on patient background and will.

This study is also a retrospective observational study, and although it is not possible to clarify the appropriate treatment for bilateral metastases, it covers more cases than similar reports in the past [17,22,23,24,25]. Comparing the OS or DSS of patients undergoing unilateral neck dissection initially and that of those undergoing bilateral dissection initially, those with bilateral dissection tended to have a worse prognosis although there was no significant difference. It seems that this is because there were more cases in which the primary lesion and metastatic lesion were advanced in the bilateral dissection group, and therefore, it is not possible to discuss the superiority or inferiority of the surgical procedure. Of the 10 patients with poor prognosis in the unilateral dissection group, two cases died of cervical metastasis, while of the 24 cases with poor prognosis in the bilateral dissection group, nine cases died of cervical metastasis. These findings suggest that initial unilateral dissection does not lead to uncontrolled contralateral neck, and that the elective dissection of the contralateral neck cannot be recommended.

This study has some limitations. First, this is a retrospective study with a small number of patients, and therefore it is difficult to generalize the results obtained. Second, the adverse events of bilateral neck dissection were not examined, and it is not possible to assess how contralateral elective dissection reduces a patient’s QOL. Third, this study could not examine the surgical procedure for contralateral neck dissection in detail due to the nature of a multicenter, retrospective study.

Since it is difficult to carry out a randomized intervention study for contralateral elective dissection, we would like to conduct observational studies using a large number of cases at multiple centers in the future and study its effectiveness and quality of life.

## 5. Conclusions

We examined oral cancer patients with pathologically proven contralateral neck metastasis. The prognosis of pN2c was poor, but not significantly different from pN2b. The most important risk factor for contralateral metastasis was T stage. OS, DSS, and neck control was not different between the first ipsilateral neck dissection group and the bilateral group. Thus, contralateral elective neck dissection is not recommended and we suggest a wait-and-see policy.

OS, DSS, and neck control of patients undergoing unilateral neck dissection initially and those undergoing bilateral dissection initially were not different, and our results advise against the upfront elective neck dissection of the contralateral neck in oral cancer in general. However, until further data are available, the decision to operate on contralateral neck can be an option, especially in large (T3/T4) tumors, involving the tongue or floor of mouth, with level IV and V involvement, and in those with clinical evidence of extranodal spread.

## Figures and Tables

**Figure 1 ijerph-19-09229-f001:**
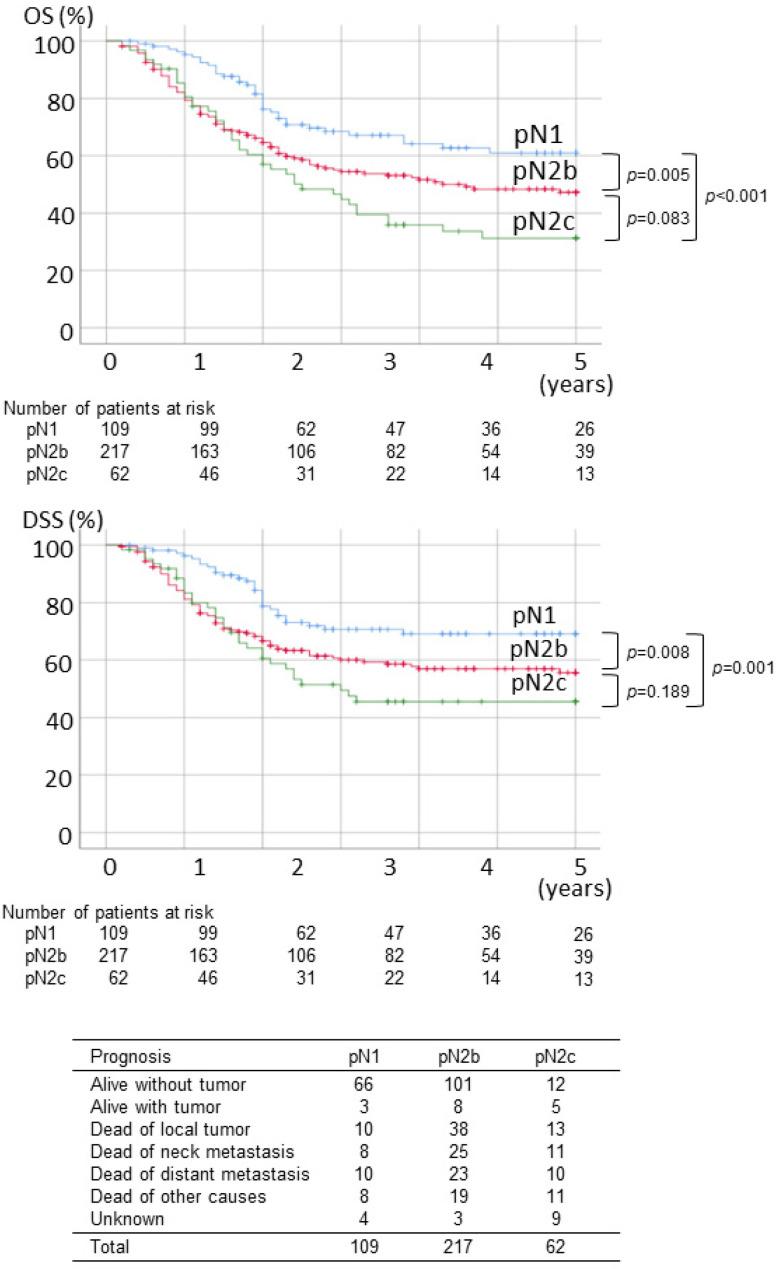
OS, DSS, and final status of patients with pN1, pN2b and pN2c neck.

**Figure 2 ijerph-19-09229-f002:**
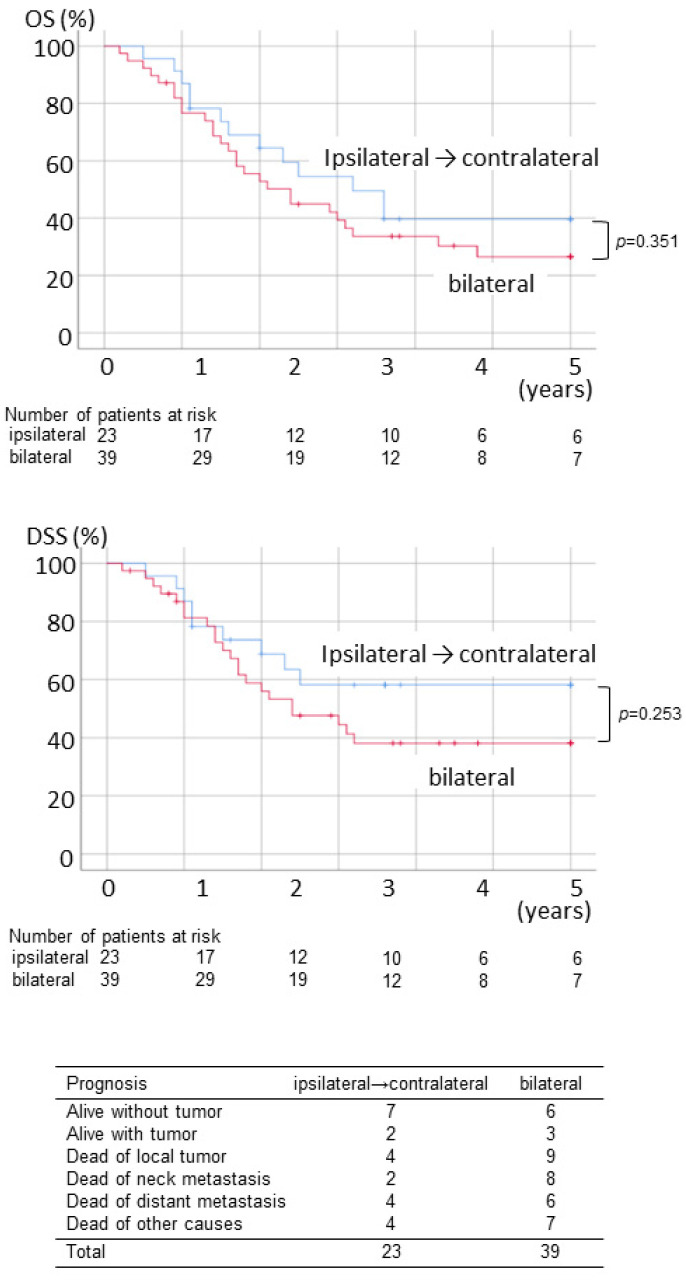
OS, DSS, and the final status of patients with pN2c necks who underwent ipsilateral neck dissection initially and bilateral neck dissection initially.

**Figure 3 ijerph-19-09229-f003:**
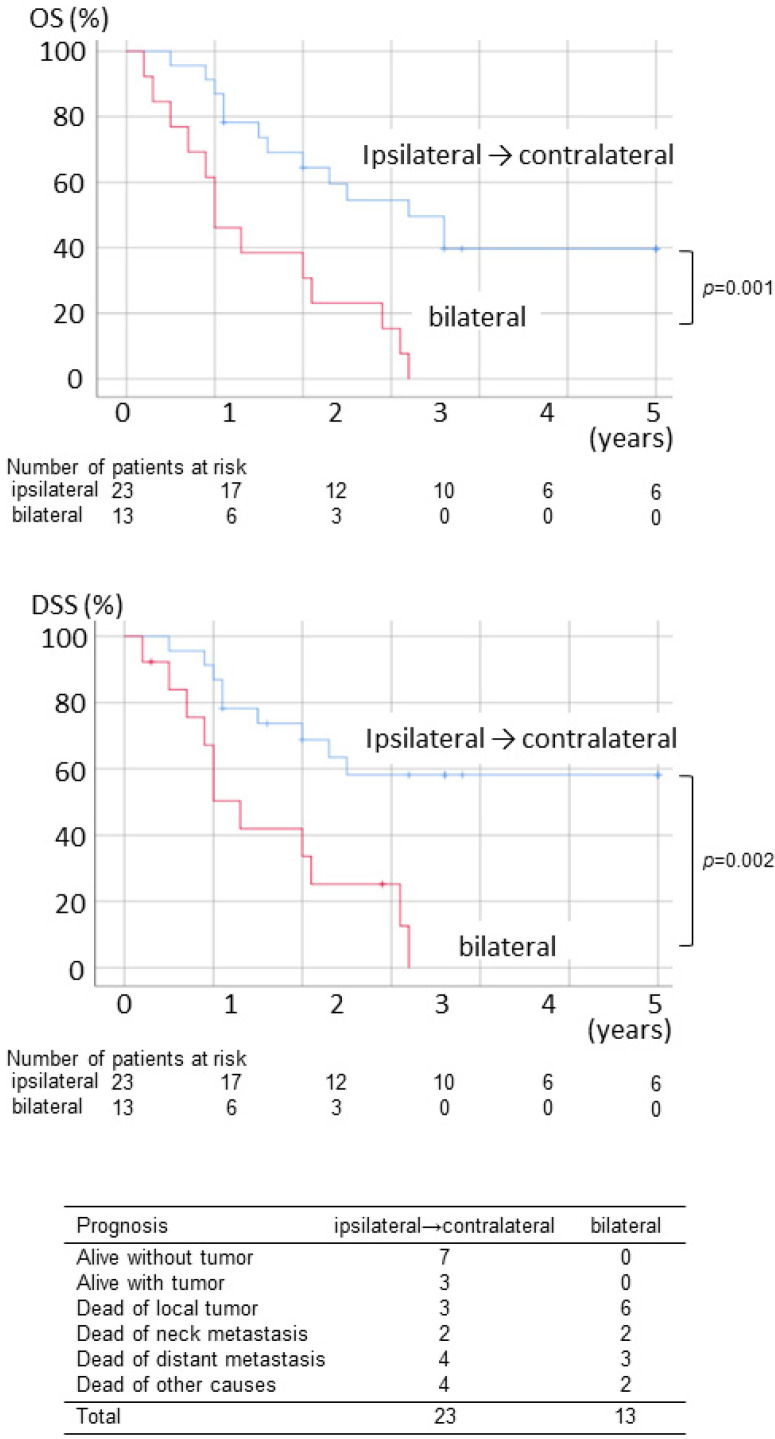
OS, DSS, and the final status of patients with pN2c necks excluding clinical N2c necks, who underwent ipsilateral neck dissection initially and bilateral neck dissection initially.

**Figure 4 ijerph-19-09229-f004:**
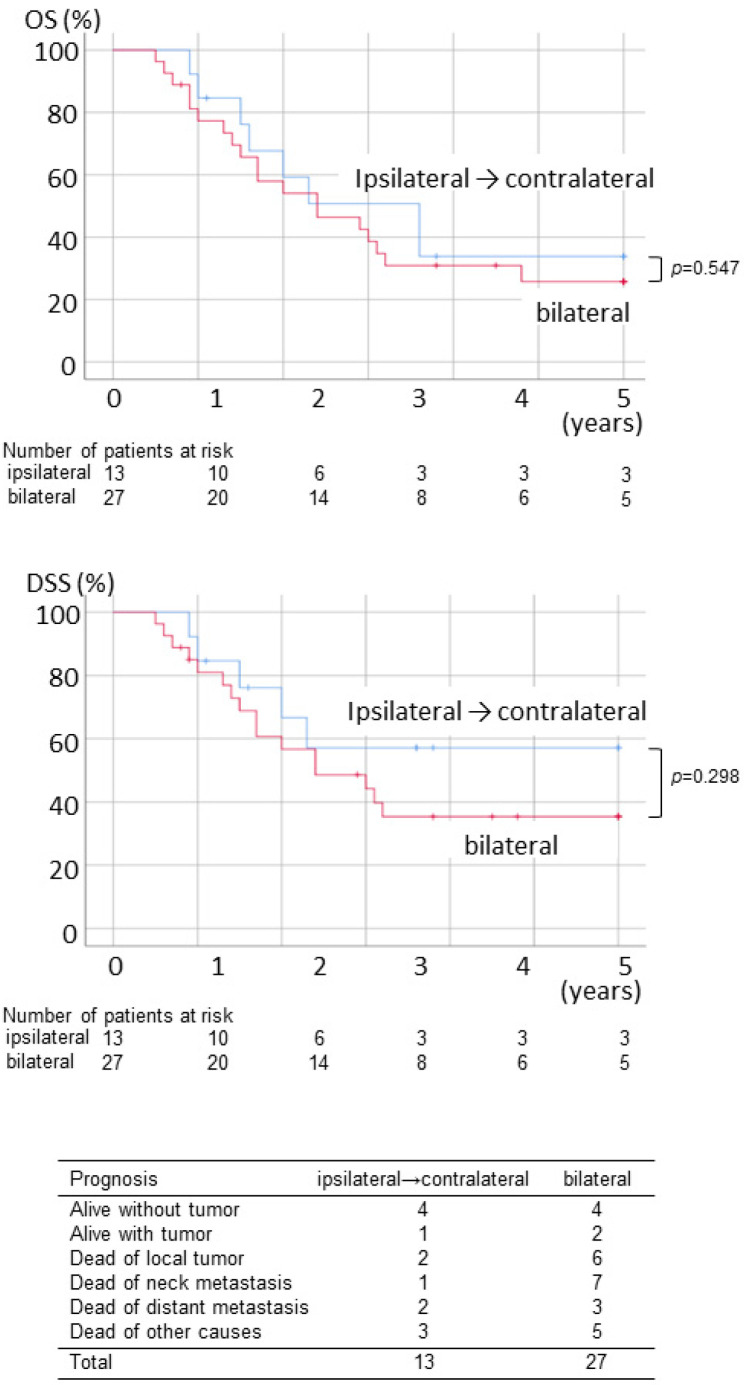
OS, DSS, and final status of the tongue or the floor of the mouth cancer patients with pN2c necks who underwent ipsilateral neck dissection initially and bilateral neck dissection initially.

**Table 1 ijerph-19-09229-t001:** Patient characteristics.

Variable		Number of Patients/Mean ± SD			
		pN1	pN2b	pN2c	Total
Age	years	64.0 ± 13.7	65.9 ± 12.0	65.8 ± 10.4	65.4 ± 12.2
Sex	male	75 (68.8%)	133 (61.3%)	48 (77.4%)	256 (66.0%)
	female	34 (31.2%)	84 (38.7%)	14 (22.6%)	132 (34.0%)
Primary site	tongue	51 (46.8%)	87 (40.1%)	25 (40.3%)	163 (42.0%)
	lower gingiva	24 (22.0%)	59 (27.2%)	8 (12.9%)	91 (23.5%)
	upper gingiva/hard palate	12 (11.0%)	32 (14.7%)	12 (19.4%)	56 (14.4%)
	floor of the mouth	12 (11.0%)	17 (7.8%)	15 (24.2%)	44 (11.3%)
	buccal mucosa	8 (7.3%)	18 (8.3%)	0 (0%)	26 (6.7%)
	others	2 (1.8%)	4 (1.8%)	2 (3.2%)	8 (2.1%)
Histological grade	well/moderately differentiated	96 (88.1%)	177 (81.6%)	56 (90.3%)	329 (84.8%)
	poorly differentiated	8 (7.3%)	31 (14.3%)	6 (9.7%)	45 (11.6%)
	unknown	5 (4.6%)	9 (4.1%)	0 (0%)	14 (3.6%)
Clinical T stage	T1	7 (6.4%)	4 (1.8%)	4 (6.5%)	15 (3.9%)
	T2	47 (43.1%)	84 (38.7%)	18 (29.0%)	149 (38.4%)
	T3	20 (18.3%)	37 (17.1%)	15 (24.2%)	72 (18.6%)
	T4a	24 (22.0%)	72 (33.2%)	24 (38.7%)	120 (30.9%)
	T4b	10 9.2%)	17 (7.8%)	1 (1.6%)	28 (7.2%)
	unknown	1 (0.9%)	3 (1.4%)	0 (0%)	4 (1.0%)
Clinical N stage	cN0	27 (24.8%)	30 (13.8%)	11 (17.7%)	68 (17.5%)
	cN1	55 (50.5%)	51 (23.5%)	6 (9.7%)	112 (28.9%)
	cN2a	0 (0%)	1 (0.5%)	1 (1.6%)	2 (0.5%)
	cN2b	20 (18.3%)	107 (49.3%)	17 (27.4%)	145 (37.4%)
	cN2c	6 (5.5%)	21 (9.7%)	26 (41.9%)	52 (13.4%)
	cN3	1 (0.9%)	7 (3.2%)	1 (1.6%)	9 (2.3%)
Total		109	217	62	388

**Table 2 ijerph-19-09229-t002:** Factors related to the prognosis of patients with pN2c necks (univariate analysis).

Variable		OS	DSS
		*p* Value	*p* Value
Age	65 or more/64 or less	0.127	0.111
Sex	male/female	0.383	0.878
Primary site	others/tongue or floor of the mouth	0.922	0.912
Tumor location	including midline/one side only	0.763	0.980
Deep invasion	invading to the midline/one side only	0.322	0.431
T stage	T3–4/T1–2	* 0.031	* 0.007
cN stage	N2–3/N0–1	0.154	0.246
Differentiation	poorly differentiated/well or moderately	0.121	0.195
Initial neck dissection	bilateral/ipsilateral	0.500	0.406
Neoadjuvant chemo-/radiotherapy	(+)/(−)	0.673	0.515
Postoperative chemo-/radiotherapy	(+)/(−)	0.968	0.787
Number of positive nodes in the ipsilateral neck	4 or more/3 or less	0.654	0.755
Extranodal spread in the ipsilateral neck	(+)/(−)	* 0.002	* 0.006
Level of metastasis in the ipsilateral neck	level 4–5/level 1–3	* 0.032	* 0.004
Number of positive nodes in the contralateral neck	4 or more/3 or less	0.768	0.281
Extranodal spread in the contralateral neck	(+)/(−)	0.053	0.429
Level of metastasis in the contralateral neck	level 4–5/level 1–3	0.974	0.579

Log rank test, * *p* < 0.05, Abbreviation: OS: overall survival; DSS: disease specific survival.

**Table 3 ijerph-19-09229-t003:** Factors related to the prognosis of patients with pN2c necks (multivariate analysis).

Variable		*p* Value	Hazard Ratio	95% Confidence Interval
(i) OS				
T stage	T3–4/T1–2	* 0.006	2.753	1.344–5.637
Extranodal spread in the ipsilateral neck	(+)/(−)	0.117	1.756	0.869–3.548
Level of metastasis in the ipsilateral neck	level 4–5/level 1–3	* 0.048	2.205	1.008–4.821
Extranodal spread in the contralateral neck	(+)/(−)	0.160	0.619	0.317–1.208
(ii) DSS				
T stage	T3–4/T1–2	* 0.003	3.883	1.569–9.609
Extranodal spread in the ipsilateral neck	(+)/(−)	0.118	1.847	0.855–3.988
Level of metastasis in the ipsilateral neck	level 4–5/level 1–3	* 0.020	2.623	1.162–3.988

Cox regression analysis, * *p* < 0.05, Abbreviation: OS: overall survival; DSS: disease specific survival.

**Table 4 ijerph-19-09229-t004:** Factors related to the prognosis of patients with pN2c necks (cancer of the tongue or the floor of the mouth, univariate analysis).

Variable		OS	DSS
		*p* Value	*p* Value
Age	65 or more/64 or less	0.552	0.378
Sex	male/female	0.453	0.214
Tumor location	including midline/one side only	0.626	0.954
Deep invasion	invading to the midline/one side only	0.264	0.247
T stage	T3–4/T1–2	0.086	* 0.034
cN stage	N2–3/N0–1	0.234	0.185
Differentiation	poorly differentiated/well or moderately	0.734	0.956
Initial neck dissection	bilateral/ipsilateral	0.547	0.298
Neoadjuvant chemo-/radiotherapy	(+)/(−)	0.599	0.722
Postoperative chemo-/radiotherapy	(+)/(−)	0.760	0.674
Number of positive nodes in the ipsilateral neck	4 or more/3 or less	0.193	0.217
Extranodal spread in the ipsilateral neck	(+)/(−)	* 0.016	0.080
Level of metastasis in the ipsilateral neck	level 4–5/level 1–3	* 0.001	* <0.001
Number of positive nodes in the contralateral neck	4 or more/3 or less	0.764	0.355
Extranodal spread in the contralateral neck	(+)/(−)	0.373	0.863
Level of metastasis in the contralateral neck	level 4–5/level 1–3	0.850	0.712

Log rank test, * *p* < 0.05, Abbreviation: OS: overall survival; DSS: disease specific survival.

**Table 5 ijerph-19-09229-t005:** Factors related to the prognosis of patients with pN2c necks (cancer of the tongue or the floor of the mouth, multivariate analysis).

Variable		*p* Value	Hazard Ratio	95% Confidence Interval
(i) OS				
T stage	T3–4/T1–2	0.244	1.597	0.727–3.508
Extranodal spread in the ipsilateral neck	(+)/(−)	0.094	2.020	0.887–4.598
Level of metastasis in the ipsilateral neck	level 4–5/level 1–3	* 0.019	3.057	1.198–7.797
(ii) DSS				
T stage	T3–4/T1–2	0.151	2.063	0.787–5.408
Extranodal spread in the ipsilateral neck	(+)/(−)	0.437	1.463	0.561–3.817
Level of metastasis in the ipsilateral neck	level 4–5/level 1–3	* 0.006	4.123	1.498–11.348

Cox regression analysis, * *p* < 0.05, Abbreviation: OS: overall survival; DSS: disease specific survival.

## Data Availability

The datasets used and analyzed during the study are available from the corresponding author upon reasonable request.

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
