# Peer review of "Treatment and Prognosis of Oral Cancer Patients with Confirmed Contralateral Neck Metastasis: A Multicenter Retrospective Analysis"

_ijerph, 2022, doi:10.3390/ijerph19159229_

Round 1

Reviewer 1 Report

The reviewer would like to thank the authors for their efforts in performing the study. Attached below are the reviewer comments.

Author Response

Thank you very much for your Reviewers’ valuable and encouraging comments which gave us a good opportunity for revising our manuscript. 

Reviewer 2 Report

This article described about lymph node metastasis in oral cancer, with a particular focus on N2c cases. It is an interesting topic because it gives answers to clinically very important questions. I think acceptance is fine after a small revision.

I have one question about the data.

The results show that the prognosis is poor when the primary is advanced. Is it possible to determine whether the primary is unilateral or bilateral? 

I would like the authors to discuss and give their opinion on whether a primary tumour on one side causes metastases on the opposite side or whether there is a bilateral primary tumour that straddles both sides, causing bilateral metastases.

Author Response

To Reviewer #2

  • The results show that the prognosis is poor when the primary is advanced. Is it possible to determine whether the primary is unilateral or bilateral? 

I would like the authors to discuss and give their opinion on whether a primary tumour on one side causes metastases on the opposite side or whether there is a bilateral primary tumour that straddles both sides, causing bilateral metastases.

(Reply)

Unfortunately, we collected tumor locations in N2c cases only, but did not investigate whether the primary tumor was beyond the midline in N1 and N2b cases. The following description was added.

Line 114-116: Regarding the location of the primary tumor in N2c patients, 15 patients had a unilateral tumor, while 47 had a primary tumor that invaded the opposite side beyond the midline.

Reviewer 3 Report

This is the review of the article entitled "Treatment and prognosis of oral cancer patients with confirmed contralateral neck metastasis: A multicenter retrospective analysis"

Abstract:

Line 33: The reviewer recommends to keep consistency using abbreviations/terms like pN2c

Line 37/38: The reviewer recommends beeing very cautious stating that there is no evidence for the recommendation of a contralateral neck dissection in pN1/2b patients. There is plenty of literature (clinical studies as well as reviews/meta-analyses showing high rates for occult lymph node metastases in pN+ patients (generally 11.5% see DOI: 10.1007/s00405-021-07230-5)

Furthermore, higher T-category, a higher number ob ipsilateral lymph node metastases and higher Grading were correlated with contralateral metastases DOI: 10.1097/00005537-200211000-00017 and DOI: 10.1002/hed.20423).

This especially applies for carcinomas near the midline and malignancies of the floor of the mouth (DOI: 10.1002/hed.20423).

Therefore, and this is generally accepted, contralateral elective neck dissection is recommended in pN+ patients.

Introduction:

Line 43,44: The number of oral cancers in the world is 18th with 2% of all cancers, and the number of deaths is 14th with 1.9% of all cancers [1]. 

Extensive english editing is recommended. The reviewer simply does not understand this sentence, because of the structure (?). This does not make any sense in its current form.

The same applies for the following sentence in lines 44 and 45.

Sentence 3 (lines 45, 46) is demonstrably false. There are 78 reviews/meta-analysis regarding the relationship between oral squamous cancer and neck dissection and over 4400 studies in general.

Line 53: Please give explanations before showing the abbreviation (QOL)

Line 56: MRI is the generally used term.

Line 58: Is 34% really rare?

Line 59: The authors state that there is no consensus of whether to perform contralateral neck dissection in patients with pN+ status, but, to the reviewers best knowledge, it is a generally accepted recommendation to do so (see f.e. treatment guide for oral squamous cell carcinoma from Germany), especially when considering the high rate of metastases in cN0 necks.

Considering the poor performance of the article, the reviewer recommends extensive english editing and much further research on the topic before resubmission. The corrections are exemplarily and the reviewer did non read further than to the M&M section.

Author Response

To Reviewer #3

  • Abstract:

Line 33: The reviewer recommends to keep consistency using abbreviations/terms like pN2c

(Reply)

Line 33: “pN2C” was corrected to “pN2c”.

  • Line 37/38: The reviewer recommends being very cautious stating that there is no evidence for the recommendation of a contralateral neck dissection in pN1/2b patients. There is plenty of literature (clinical studies as well as reviews/meta-analyses showing high rates for occult lymph node metastases in pN+ patients (generally 11.5% see DOI: 10.1007/s00405-021-07230-5)

Furthermore, higher T-category, a higher number of ipsilateral lymph node metastases and higher Grading were correlated with contralateral metastases DOI: 10.1097/00005537-200211000-00017 and DOI: 10.1002/hed.20423).

This especially applies for carcinomas near the midline and malignancies of the floor of the mouth (DOI: 10.1002/hed.20423).

Therefore, and this is generally accepted, contralateral elective neck dissection is recommended in pN+ patients.

Line 37-39: “Therefore, contralateral neck dissection needs be performed only when patients have clinically positive metastases in the contralateral necks.” was revised to “Therefore, the indication for contralateral elective neck dissection in N1/2b patients should be carefully determined in consideration of individual conditions.”.

  • Introduction:

Line 43,44: The number of oral cancers in the world is 18th with 2% of all cancers, and the number of deaths is 14th with 1.9% of all cancers [1].

Extensive english editing is recommended. The reviewer simply does not understand this sentence, because of the structure (?). This does not make any sense in its current form.

(Reply)

Line 43-44: “The number of oral cancers in the world is 18th with 2% of all cancers, and the number of deaths is 14th with 1.9% of all cancers [1]” was revised to “The number of oral cancers in the world is 18th, 2% of all cancers, and the number of deaths is 14th, 1.9% of all cancers [1]”.

  • The same applies for the following sentence in lines 44 and 45.

Sentence 3 (lines 45, 46) is demonstrably false. There are 78 reviews/meta-analysis regarding the relationship between oral squamous cancer and neck dissection and over 4400 studies in general.

(Reply)

Line 45-47: “Therefore, there are few clinical studies with a high level of evidence about oral cancer, and the current situation is that treatment methods vary on each hospital.” was revised to “Therefore, there are not many studies with a high level of evidence using a large number of cases of oral cancer, and the current situation is that treatment methods vary on each hospital.”.

  • Line 53: Please give explanations before showing the abbreviation (QOL)

(Reply)

Line 53: “QOL” was revised to “quality of life (QOL)”.

  • Line 56: MRI is the generally used term.

(Reply)

Line 56: “MR” was revised to “MRI”.

  • Line 58: Is 34% really rare?

(Reply)

Line 58-59: “Contralateral or bilateral neck metastasis of oral cancer is rare, 0.9-34.7% [8,10–12].” was revised to “Some reports suggest that the frequency of contralateral or bilateral neck metastasis of oral cancer is as rare as 0.9% and others as high as 24% [8, 10-12].

  • Line 59: The authors state that there is no consensus of whether to perform contralateral neck dissection in patients with pN+ status, but, to the reviewers best knowledge, it is a generally accepted recommendation to do so (see f.e. treatment guide for oral squamous cell carcinoma from Germany), especially when considering the high rate of metastases in cN0 necks.

(Reply)

The concept of elective neck dissection for oral cancer is slightly different in Japan. Elective dissection on the ipsilateral neck for N0 and that on the contralateral neck for N1/2b may be performed depending on the location and size of the primary tumor resection and the reconstructive surgery, but in general, many oral oncologists follow a wait-and-see policy.

Lines 216-226 provide further details.

Reviewer 4 Report

The authors visited the treatment and prognosis of pN2c oral cancer based on data from 62 pN2c patients out of 388 pN+ patients with oral squamous cell carcinomas. Statistical analysis was performed on the various factors with overall survival and disease-specific survival. The authors conducted a multivariate cox regression analysis and reported the results with corresponding significant levels and confidence intervals.

The statistical analysis in the manuscript must be improved or modified. I appreciate the fact that the authors reported p-values and confidence intervals in all the Tables and Figures. Unfortunately, according to the manuscript, no model diagnosis was performed to evaluate the validation of assumptions for the adopted methods. Those assumptions are critical to justify whether the produced p-values are usable or not. In case the assumptions are violated, the produced-p-values would be non-sense and most of the conclusions in the manuscript would not be supported by the data.

The authors must check the assumptions based on their model and make appropriate remedies. In case certain violations are non-fixable, the authors should report them and not choose to use results that require the corresponding assumptions.

If the authors choose not to evaluate the assumptions, then all p-values, confidence intervals, and their related conclusions must be removed from the manuscript. The scientific soundness will be severely impacted if they were removed.

Author Response

To Reviewer #4

The statistical analysis in the manuscript must be improved or modified. I appreciate the fact that the authors reported p-values and confidence intervals in all the Tables and Figures. Unfortunately, according to the manuscript, no model diagnosis was performed to evaluate the validation of assumptions for the adopted methods. Those assumptions are critical to justify whether the produced p-values are usable or not. In case the assumptions are violated, the produced-p-values would be non-sense and most of the conclusions in the manuscript would not be supported by the data.

The authors must check the assumptions based on their model and make appropriate remedies. In case certain violations are non-fixable, the authors should report them and not choose to use results that require the corresponding assumptions.

If the authors choose not to evaluate the assumptions, then all p-values, confidence intervals, and their related conclusions must be removed from the manuscript. The scientific soundness will be severely impacted if they were removed.

(Reply)

Model diagnosis was performed in each cox proportional hazard model, and the following revisions were made.

Line 105-106: The description “The goodness of fit of the proportional hazards analysis was diagnosed using SPSS to obtain log minus log and DfBeta.” was added.

Line 144-150:

“Table 3 shows multivariate Cox regression analysis of prognostic factors. Advanced T stage were significantly correlated with poor OS (HR, 2.562; 95% CI, 1.238-5.302; p=0.011), and with poor DSS (HR, 2.558; 95% CI, 1.137-5.755; p=0.023).” was revised to “Table 3 shows multivariate Cox regression analysis of prognostic factors. Advanced T stage was significantly correlated with poor OS (HR, 2.468; 95% CI, 1.174-5.191; p=0.017), and with poor DSS (HR, 4.005; 95% CI, 1.526-10.514; p=0.005). Furthermore, extranodular spread in the ipsilateral neck was significantly correlated with poor OS (HR, 2.114; 95% CI, 1.083-4.124; p=0.028), and level 4-5 metastasis significantly correlated with poor DSS (HR, 2.628; 95% CI, 1.163-5.941; p=0.020. The timing of surgery of the contralateral neck did not correlated with survival also by multivariate analysis.”.

Line 157: The following description was deleted.

3.4. Prognosis by Time of Surgery for Contralateral Metastasis

There were differences in the background factors between pN2c patients under-going ipsilateral neck dissection initially (23 patients) and bilateral neck dissection (39 patients). Significant differences between the two groups were found in cT stage (p=0.013), cN stage (p≺0.001), neoadjuvant chemo-/radiotherapy (p=0.002), and number of positive nodes in the ipsilateral neck (p=0.003), which indicated that those undergoing bilateral dissection initially were more advanced cases (Table 4). Factors that were sig-nificant between these two groups were added to the covariates, and the relationship between the initial neck dissection method and prognosis was analyzed by multivariate analysis. As a result, the factor significantly correlated with prognosis was T stage, and the initial neck dissection method was not associated with OS and DSS (Table 5).

Table 3 was revised.

The Cox regression analysis was redone with the factors that were significant in the univariate analysis and the initial cervical dissection method added to the covariates. We made a diagnosis of model suitability and added the omnibus test of model coefficients to the table.

Tables 4 and 5 were deleted.

Table 6 was changed to Table 4.

Table 7 was revised and changed to Table 5.

Round 2

Reviewer 1 Report

The reviewer would like to thank the authors for revising the manuscript. Please correct the reference in line 59 to be 12-19 not 19-12. Thank you!

Author Response

Line 59: “19-12” was corrected to “12-19”.

Reviewer 4 Report

The authors did not get my point in the revision. The Cox regression model assumes that the effects of the predictor variables upon survival are constant over time and are additive on one scale. The authors must evaluate the assumptions, otherwise, the reported p-values, confidence intervals, and corresponding results are non-sense if the assumption is failed.

Author Response

To Reviewer #4

The authors did not get my point in the revision. The Cox regression model assumes that the effects of the predictor variables upon survival are constant over time and are additive on one scale. The authors must evaluate the assumptions, otherwise, the reported p-values, confidence intervals, and corresponding results are non-sense if the assumption is failed.

(Reply) Thank you for your very meaningful comments. We verified the proportional hazard nature of the results and made some corrections to the table of results.

Also, log minus log curves are shown for reference.

  1. Line 103-106: The description was revised as follows;

Factors related to prognosis were analyzed by Cox proportional hazards model using variables with a p value less than 0.1 in the log-rank test as covariates. Proportional hazardousness was determined by drawing a log-minus-log curve.

  1. Line 132-144: The first and second paragraphs were revised as follows;

Table 2 shows the results of log-rank test of factors related to the prognosis of pN2C. OS was significantly lower in patients with advanced T stage (p=0.031), extranodal spread in the ipsilateral neck (p=0.002), and level 4-5 metastasis in the ipsilateral neck (p=0.032). DSS was significantly lower in those with advanced T stage (p=0.007), extranodal spread in the ipsilateral neck (p=0.006), and level 4-5 metastasis in the ipsilateral neck (p=0.004). However, the timing of surgery of the contralateral neck (initially ipsilateral and consequently contralateral neck / initially bilateral necks) did not become a significant factor related to the prognosis of pN2c (OS: p=0.500, DSS: p= 0.406).

Table 3 shows multivariate Cox regression analysis of prognostic factors. Advanced T stage was significantly correlated with poor OS (HR, 2.753; 95% CI, 1.344-5.1637; p=0.006), and with poor DSS (HR, 3.883; 95% CI, 1.569-9.609; p=0.00). Furthermore, level 4-5 metastasis significantly correlated with poor OS (HR, 2.205; 95% CI, 1.008-4.821; p=0.048), and DSS (HR, 2.623; 95% CI, 1.162-53.988; p=0.020).

  1. Tables 2-5 were revised.
